# COVID-19 and Postural Control—A Stabilographic Study Using Rambling-Trembling Decomposition Method

**DOI:** 10.3390/medicina58020305

**Published:** 2022-02-17

**Authors:** Magdalena Żychowska, Kamila Jaworecka, Ewelina Mazur, Kajetan Słomka, Wojciech Marszałek, Marian Rzepko, Wojciech Czarny, Adam Reich

**Affiliations:** 1Department of Dermatology, Institute of Medical Sciences, Medical College of Rzeszow University, 35-055 Rzeszów, Poland; magda.zychowska@gmail.com (M.Ż.); kamila.jaworecka@gmail.com (K.J.); mazur.eveline@gmail.com (E.M.); 2Institute of Sport Sciences, The Jerzy Kukuczka Academy of Physical Education, 40-065 Katowice, Poland; k.slomka@awf.katowice.pl (K.S.); w.marszalek@awf.katowice.pl (W.M.); 3Institute of Physical Culture Sciences, Medical College of Rzeszow University, 35-326 Rzeszów, Poland; marianrzepko@poczta.onet.pl (M.R.); wojciechczarny@wp.pl (W.C.)

**Keywords:** SARS-CoV-2, stabilometry, postural control, stabilography, COVID-19, rambling-trembling

## Abstract

*Background* *and Objectives*: Some respiratory viruses demonstrate neurotropic capacities. Severe acute respiratory syndrome coronavirus 2 (SARS-CoV-2) has recently taken over the globe, causing coronavirus disease 2019 (COVID-19). The aim of the study was to evaluate the impact of COVID-19 on postural control in subjects who have recently recovered from the infection. *Materials and Methods*: Thirty-three convalescents who underwent COVID-19 within the preceding 2–4 weeks, and 35 healthy controls were enrolled. The ground reaction forces were registered with the use of a force platform during quiet standing. The analysis of the resultant center of foot pressure (COP) decomposed into rambling (RAMB) and trembling (TREMB) and sample entropy was conducted. *Results*: Range of TREMB was significantly increased in subjects who experienced anosmia/hyposmia during COVID-19 when the measurement was performed with closed eyes (*p* = 0.03). In addition, subjects who reported dyspnea during COVID-19 demonstrated significant increase of length and velocity of COP (*p* < 0.001), RAMB (*p* < 0.001), and TREMB (*p* < 0.001), indicating substantial changes in postural control. *Conclusions*: Subjects who had experienced olfactory dysfunction or respiratory distress during COVID-19 demonstrate symptoms of balance deficits after COVID-19 recovery, and the analysis using rambling-trembling decomposition method might point at less efficient peripheral control. Monitoring for neurological sequelae of COVID-19 should be considered.

## 1. Introduction

Some respiratory viruses, including coronaviruses, demonstrate neurotropic capacities and ability to trigger immune response in the nervous system in vulnerable populations [1]. The neurological manifestations are usually related to the “cytokine storm,” which results from the immune reaction to the infection of the central nervous system (CNS). Neuroinvasive properties of severe acute respiratory syndrome coronavirus (SARS-CoV) have already been demonstrated in human and animal models [2]. SARS-CoV-2, a virus structurally similar to SARS-CoV, has recently taken over the globe, causing coronavirus disease 2019 (COVID-19) pandemic. Neurological manifestations have also been reported in the course of COVID-19 and included encephalopathy, cerebrovascular events, encephalitis, acute myelitis, and Guillain–Barre syndrome [1]. Some authors also highlight potential persistence of coronaviruses in the nervous system [3]. Long-term complications of SARS-CoV-2 infection are increasingly recognized and have recently become major concerns. These sequelae include not only pulmonary fibrosis but also direct or indirect damage to the neurons in the CNS or peripheral nervous system (PNS).

Clinical measurements of balance, including Berg Balance Scale, Timed Up and Go test (TUG), Tinetti test, and Functional Reach Test (FRT), are commonly used in everyday practice to assess balance, walking ability, limits of stability, and fall risk. However, these tests often fail to detect discrete balance deficits in early stages of neurological conditions [4]. The evaluation of the static balance may be also performed in an objective, non-invasive, and quick way by monitoring the migration of center of pressure (COP) by means of force platforms [5,6]. The main limitation of assessments based only on COP analysis is the fact that numerous components have impact on balance control, including sensory systems, dynamic control of muscles, and passive properties of musculoskeletal system and ligaments [7]. In our research, we used the rambling-trembling decomposition method to assess the COP trajectory in subjects who underwent SARS-CoV-2 infection. This method was developed by Zatsiorsky and Duarte, who distinguished two components making up the stabilogram [8]. The trembling component (TREMB) reveals body oscillation around reference point trajectory, while the rambling component (RAMB) reflects the movement of a reference point with respect to which the balance of the body is promptly kept [9]. Prior studies demonstrated that RAMB reflects predominantly the contribution of the supraspinal centers to the postural control and the processes of the CNS, while TREMB trajectory is largely dependent on peripheral components, including spinal reflexes and mechanical properties of the musculoskeletal system [10,11]. Some authors also postulated that TREMB reflects the differences between motor planning and output [11].

The aim of the current study was to evaluate the impact of COVID-19 on postural control in subjects who have recently recovered from the infection. We speculated that SARS-CoV-2 infection might have impacted the neuromuscular control, leading to postural control problems in COVID-19 convalescents. We used dynamic measures of postural control that might be sensitive enough to reveal the subtle neuromuscular changes in the subjects.

## 2. Material and Methods

A case-control study was carried out between November 2020 and January 2021 to compare stabilographic parameters characterizing postural control in otherwise healthy adult subjects who had recently recovered from COVID-19 (study group) and age- and sex-matched healthy volunteers who had not been infected with SARS-CoV-2 (control group). The study was approved by the local ethics committee (Ethics Committee of the University of Rzeszow, decision No 6/11/2020) and was conducted in accordance with the Helsinki Declaration. All participants signed the written informed consent before inclusion in the study.

### 2.1. Participants

Thirty-three patients (6 men and 27 women, mean age 40.0 ± 12.8 years) who underwent COVID-19 infection within the preceding 2–4 weeks and 35 healthy controls (9 men and 26 women, mean age 38.9 ± 14.4 years) were enrolled in the study. SARS-CoV-2 infection was confirmed in each case using a polymerase chain reaction (PCR) test. Subjects in the control group did not report any recent symptoms of infection and were negative for anti-SARS-CoV-2 IgM and IgG antibodies. Both groups were similar in terms of basic somatic variables, including age (*p* = 0.7), gender (*p* = 0.45), body mass (*p* = 0.99), and height (*p* = 0.75).

None of the study participants were competitive athletes, took any concomitant medications, or suffered from any chronic condition that might impact the postural stability. Subjects who reported balance deficits or vestibular disorders or had a history of neurological impairments were excluded from the study. Detailed medical history regarding the course of COVID-19 was taken with each participant in the study group.

### 2.2. Stabilographic Measurements

A force platform (AMTI, AccuGait, Watertown, MA, USA) was used for evaluation of the COP trajectories. The force platform registered six components of postural dynamics. Three of them were ground reaction forces, namely mediolateral force (Fx), anteroposterior force (Fy), and vertical force (Fz), and the remaining three were moment components determined around the respective axes: Mx, My, and Mz. Digital output from the AMTI platform was recorded by means of AMTI’s Netforce software at the frequency of 100 Hz. Each study participant performed six trials of quit standing, with the arms along the sides and gaze focused on a target on the wall in front of the subject in the distance of 2 m. First, 3 out of 6 trials were performed with opened eyes, and the subsequent 3 trials with closed eyes. Each time, the subject was asked to stand barefoot in a comfortable foot position on the force platform for a period of 30 s. The subject was instructed to step off the force platform between the trials and rest for 60 s to avoid fatigue. The experiment was conducted in a closed room to minimize noise or other disturbances.

The raw force platform data were further processed in MATLAB software (Mathworks Inc., Natick, MA, USA) using low-pass-4th-order Butterworth filter with a cut-off frequency of 7 Hz.

COP is a reflection of the system’s neuromuscular response to the imbalances of the body’s center of gravity. It was calculated in anterior-posterior (AP) and medio-lateral (ML) directions, using the following formulas:(1)$$\begin{array}{c} COP_{AP}=\frac{M_{x}-\left(Zoff*F_{y}\right)}{F_{z}}\\ COP_{ML}=\frac{-(M_{y}+\left(Zoff*F_{x}\right))}{F_{z}} \end{array}$$

(*M_x_* and *M_y_* are the moments around the frontal and sagittal axis, respectively; *Fx* and *Fy* are the horizontal components of the ground reaction force in ML and AP directions, respectively; *Fz* is the vertical component of ground reaction force; *Zoff* is the vertical offset from the top plate to the origin of the force platform).

Zatsiorsky and Duarte’s method [8] for rambling and trembling decomposition was used.

The parameters of the COP and signals obtained after its decomposition, namely rambling (RAMB) and trembling (TREMB), were calculated separately for tests with opened and closed eyes:

Sway ranges of COP, RAMB, and TREMB (raCOP, raRAMB, raTREMB) indicated the maximum excursion in cm;

Root mean square of COP, RAMB, and TREMB (rmsCOP, rmsRAMB, rmsTREMB) indicated the displacements around the mean value (variability of the excursion in cm);

The length of the COP, RAMB, and TREMB paths (lenCOP, lenRAMB, lenTREMB);

Mean velocity of COP, RAMB, and TREMB (vCOP, vRAMB, vTREMB), which was calculated by dividing the total length of the trajectory in cm by the recording time length in seconds.

In addition, the entropy of the COP signal (sample entropy) was calculated using Richman and Moorman’s algorithm [12]. Sample entropy is defined as negative logarithm of the probability that a dataset of length *N* that have repeated itself for *m* samples within a tolerance *r* will repeat itself for *m* + 1 samples (without self-matches). For each calculation, parameters *m* and *r* are fixed, and *m* is the length of sequences to be compared, and *r* is the tolerance for accepting matches. The recommended values for the parameters (*m* = 3 and *r* = 0.2) were used in current study.

All calculations for COP, RAMB, and TREMB were performed in AP (sagittal plane) and ML (frontal plane) directions.

### 2.3. Statistical Analysis

Statistical analysis was performed using Statistica^®^ 13.0 (TIBCO Software Inc., Kraków, Poland). Numerical data are expressed as mean with standard deviation (SD) of the mean and median. Categorical data are presented as numbers and percentages. Chi-square test was used to compare categorical data. As the majority of parameters did not show normal distribution according to Shapiro–Wilk test, the Mann–Whitney U test was used for comparisons between groups. The *p*-value < 0.05 was considered to be statistically significant.

## 3. Results

### 3.1. The Course of COVID-19

Anosmia or partial loss of smell (hyposmia) was the most common symptom of COVID-19 and was reported by 24 (72.7%) subjects in the study group. Six (18.2%) patients suffered from shortness of breath during the infection. None of the subjects experienced severe course of the disease or were hospitalized due to COVID-19. Detailed demographic and clinical characteristics of study participants are presented in Table 1.

### 3.2. Postural Control in Subjects Who Underwent COVID-19 versus Healthy Controls

The spatiotemporal parameters of the COP (raCOP, rmsCOP, lenCOP, vCOP, entropy); RAMB (raRAMB, rmsRAMB, lenRAMB, vRAMB); and TREMB (raTREMB, rmsTREMB, lenTREMB, vTREMB) in the sagittal (AP) and frontal (ML) plane, for the trials performed with opened and closed eyes, were compared between the study and control group. No statistically significant differences in any of the stabilographic parameters were found between subjects who underwent COVID-19 and healthy controls (*p* > 0.05). COP migrations in subjects who recovered from COVID-19 and healthy controls are presented in Figure 1.

For further analysis, the study group was divided into subgroups depending on the presence of selected symptoms of COVID-19.

### 3.3. Postural Control in Subjects with Olfactory Abnormalities

To evaluate if there are any balance disturbances in patients who had olfactory abnormalities during the SARS-CoV-2 infection, we compared the stabilographic parameters between subjects who reported anosmia/hyposmia during COVID-19 (*n* = 24) and those who did not (*n* = 9) have olfactory abnormalities. There were no statistically significant differences in sagittal and frontal plane for the measurements with opened eyes (*p* > 0.05). However, raTREMB and rmsTREMB for sagittal plane were significantly increased in subjects with olfactory abnormalities when their eyes were closed (*p* = 0.03 and *p* = 0.04, respectively). COP migrations in subjects who suffered from olfactory abnormalities during COVID-19 and those who did not report such ailments are presented in Figure 2.

### 3.4. Postural Control in Subjects with Dyspnoea

In the next step, we compared the stabilographic parameters in convalescents with positive history of dyspnea and those who did not report such symptom during the course of SARS-CoV-2 infection. Several statistically significant differences were noted in the stabilographic parameters between the two groups. The differences were observed for assessments with opened and closed eyes but were almost exclusively present in the sagittal plane. For measurements with opened eyes, lenCOP (*p* = 0.042), vCOP (*p* = 0.042), lenRAMB (*p* = 0.038), vRAMB (*p* = 0.038), raTREMB (*p* = 0.03), rmsTREMB (*p* = 0.02), lenTREMB (*p* = 0.042), and vTREMB (*p* = 0.042) were significantly increased in the sagittal plane in subjects with dyspnea. For trials with closed eyes, lenCOP (*p* < 0.001), vCOP (*p* < 0.001), lenRAMB (*p* < 0.001), vRAMB (*p* < 0.001), raTREMB (*p* = 0.002), rmsTREMB (*p* = 0.002), lenTREMB (*p* = 0.002), and vTREMB (*p* = 0.002) as well as entropy of COP (*p* = 0.038) were significantly increased in the sagittal plane in subjects with respiratory problems during the course of SARS-CoV-2 infection. In addition, raTREMB and rmsTREMS were also significantly increased in the frontal plane (ML direction) for trials with closed eyes in patients who suffered from dyspnea (*p* = 0.047 and *p* = 0.042, respectively). Details are presented in Table 2 and Table 3. COP migrations in subjects who suffered from dyspnea during COVID-19 and those who did not report this symptom are demonstrated in Figure 3.

### 3.5. Stabilographic Measurements and Other Symptoms of COVID-19

Further subgroup analysis considering various clinical symptoms of SARS-CoV-2 infection (Table 1) did not reveal any further significant differences regarding the stabilographic parameters between convalescents with and without particular ailments (data not shown).

## 4. Discussion

Post-COVID-19, also referred to as chronic COVID-19, is increasingly recognized in patients who were initially believed to have completely recovered from the infection. It may result in gradual loss of lung function and long-term damage of the CNS and PNS [13,14]. One of the major concerns is also the putative increased risk of development of neurodegenerative conditions, including Alzheimer’s disease, Parkinson’s disease, and multiple sclerosis, following the SARS-CoV-2 infection.

Respiratory viruses may enter the nervous system through hematogenous route, e.g., by infecting the cells of the circulatory system and crossing the blood-brain barrier (BBB), or through neuronal retrograde routes by infecting directly nerve endings and using the axonal transport to reach the CNS [15]. Several recent findings favor the neurotropic and neuroinvasive capacities of SARS-CoV-2. The receptor that binds to S protein, ACE2, is highly expressed in the endothelial cells of the brain microvasculature [16,17]. Immune cells, e.g., macrophages, also express ACE2 and may carry SARS-CoV-2 inside their cytoplasm across the BBB [18]. In addition, high expression of ACE2 in various sites of the brain constitutes potential target for SARS-CoV-2 and may not only facilitate acute brain damage, but it may also provide a basis for long-term neurodegeneration [19].

The glial cells constitute the first line of defense once the virus enters the CNS, and the activated microglia are considered a marker of neuropathology and neuroinflammation [20]. The virus itself may trigger neurodegeneration either directly using its cytolytic effects or indirectly through secondary immune responses [21]. Some authors postulate that the alterations in the expression of proteins responsible for axonal transport and synaptic transmission that occur during the viral infection are similar to the changes observed in early neurodegenerative conditions [22,23,24]. SARS-CoV-2 infection may accelerate processes such as neuroinflammation, synaptic pruning, and neuronal loss, which also constitute the structural basis of Alzheimer’s disease [25]. Moreover, SARS-CoV-2 has recently been shown to cause demyelination of the brain and spinal cord and has been linked to signs similar to those of multiple sclerosis [26,27].

Some respiratory viruses, including SARS-CoV-1 and MERS-CoV, may reach the brain using the olfactory nerve [16,28]. Anosmia or partial loss of smell are commonly reported in association with SARS-CoV-2 infection. Olfactory dysfunction in patients with COVID-19 is suggested to result from direct damage of the receptor neurons in the olfactory epithelium. One of the potential theories is that SARS-CoV-2 in the initial phase enters high-ACE2-expressing nonneuronal cells in the olfactory epithelium and subsequently passes to low-ACE2-expressing neurons, from where it is anterogradely transported along axons to the brain [1]. In addition, sustentacular cells and stem cells in the olfactory epithelium widely express transmembrane serine protease 2 (TMPRSS2), another binding protein for SARS-CoV-2 [17].

Some authors also hypothesize that SARS-CoV-2 may enter the CNS through the vagus nerve and subsequently invade the respiratory control center located in close proximity to the vagal nucleus in the brainstem. As a result, the respiratory distress related to the infection in the lungs may be aggravated when the virus reaches the brainstem [1,29].

The current study aimed at assessing the postural control by means of rambling-trembling stabilogram decomposition in patients who had recently recovered from SARS-CoV-2 infection. The rambling-trembling method has already been successfully utilized to evaluate the stabilographic parameters in various musculoskeletal and neurological conditions and has proven to be reliable [30,31,32]. As there is growing amounts of evidence that SARS-CoV-2 may invade both CNS and PNS and has the potential for long-term sequalae, we hypothesized that examining patients who had recently recovered from COVID-19 by means of a static posturography may help detecting discrete disturbances in postural control and potentially identify the origin of the abnormalities (supraspinal or spinal/peripheral level). The first part of the analysis was to compare postural control in subjects who had undergone COVID-19 and in healthy individuals. Although the spatiotemporal parameters of the COP, RAMB, and TREMB in sagittal and frontal plane, both for trials with opened and closed eyes, were increased in subjects who had COVID-19, the differences did not reach statistical significance for any of the stabilographic measures. In the second part, we focused on symptoms during SARS-CoV-2 infections and found that anosmia/hyposmia and respiratory distress may be potentially related to subsequent problems in postural control as an indicator of nervous system involvement. A significantly greater value of tested parameters that might indicate impaired stability were observed in subjects who reported olfactory or respiratory abnormalities. The differences were observed nearly exclusively in the sagittal plane and were more pronounced in assessments with closed eyes. Subjects who reported olfactory abnormalities during COVID-19 demonstrated significantly increased sway range of TREMB and root mean square of TREMB in sagittal plane when compared to subjects without this symptom. In patients who had respiratory problems during the SARS-CoV-2 infection, the length and mean velocity of COP and its components (RAMB and TREMB) as well as the sway range and root mean square of TREMB were significantly increased in sagittal plane in measurements with opened eyes. All these differences were even more pronounced during assessments with closed eyes. In a recent study, Degani et al. [33] reported similar results in early stages of ageing, which were attributed, among others, to the progressive age-related changes in modulation of spinal reflex gains. This might suggest that COVID-19 elicits similar neurodegenerative changes as ageing. In addition, evaluation with closed eyes revealed significantly increased entropy of COP in the sagittal plane and range and root mean square of TREMB in the frontal plane. Sample entropy reflects the regularity of the COP signal. Higher values of entropy noted in subjects with respiratory problems correspond to more irregular, random signals, while lower values observed in subjects without dyspnea mean that the COP signal was more regular. Similar to our results, prior studies using multiscale entropy and sample entropy demonstrated higher entropy for the COP in AP in older adults but not significant differences in the COP in ML plane [34,35]. This might be alarming since a recent study reported a link between changes in postural sway complexity at multiple scales and higher fall rates in older adults [36]. Although the subgroups of patients were relatively small to draw reliable conclusions, the results point at abnormalities mainly at the peripheral/spinal level in subjects who experienced anosmia/hyposmia or breathing problem in the course of COVID-19. Significantly increased sway range of the trembling trajectory may indicate less efficient spinal control of posture stability in these two subgroups.

A major limitation of the study is a small sample size, particularly in the subgroup analyses. It is also worth noting that all subjects in the study group had mild-to-moderate course of COVID-19 and did not require treatment in hospital settings. Therefore, results of the current study may be influenced by the number of participants and disease severity. Another limitation is the lack of clinical balance tests and additional investigation of the activity of CNS and PNS. Electromyographic analysis might have provided valuable information about the bioelectric activity of muscles engaged in postural stability. Undoubtedly, further larger studies are required in order to reliably assess the postural control abnormalities after COVID-19 and to evaluate whether the rambling-trembling decomposition method may be indeed a good screening tool for rapid identification of subjects with neurological changes at the supraspinal or spinal level in the course of the SARS-CoV-2 infection.

## 5. Conclusions

Coronaviruses were found to have neurotropic potential and to cause neurological complications. The current study demonstrated impaired postural control in subjects who experienced olfactory dysfunction or respiratory distress during COVID-19, and the analysis using rambling-trembling decomposition method might point at less efficient peripheral/spinal control of stability in these patients. Timely detection of neurological deficits in patients who recover from COVID-19 might enable early implementation of therapeutic management. Monitoring for short- and long-term complications of SARS-CoV-2 infection should be recommended.

## Figures and Tables

**Figure 1 medicina-58-00305-f001:**
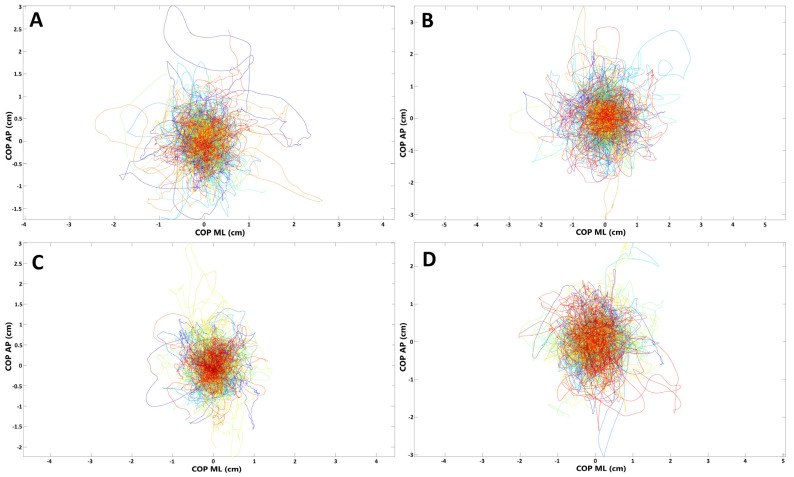
Presentation of COP (center of pressure) migrations in (**A**,**B**) subjects who recovered from COVID-19; (**C**,**D**) healthy controls (each line represents single participant; AP, anterior–posterior (frontal) plane; ML, medial–lateral (sagittal) plane). (**A**,**C**) Trials with eyes open; (**B**,**D**) trials with eyes closed.

**Figure 2 medicina-58-00305-f002:**
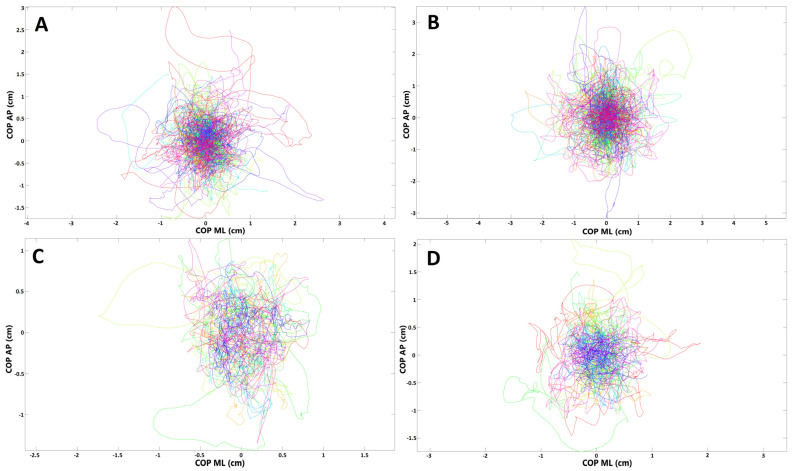
Presentation of COP (center of pressure) migrations in (**A**,**B**) subjects who had olfactory abnormalities during COVID-19; (**C**,**D**) subjects without such ailments (each line represents single participant; AP, anterior–posterior (frontal) plane; ML, medial–lateral (sagittal) plane). (**A**,**C**) Trials with eyes open; (**B**,**D**) trials with eyes closed.

**Figure 3 medicina-58-00305-f003:**
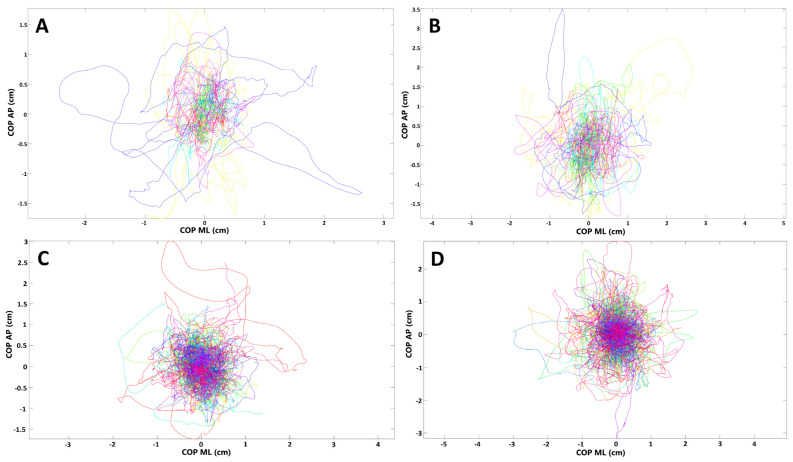
Presentation of COP (center of pressure) migrations in (**A**,**B**) subjects who had respiratory distress during COVID-19; (**C**,**D**) subjects without such ailments (each line represents single participant; AP, anterior–posterior (frontal) plane; ML, medial–lateral (sagittal) plane). (**A**,**C**) Trials with eyes open; (**B**,**D**) trials with eyes closed.

**Table 1 medicina-58-00305-t001:** Clinical characteristics of study participants (SD, standard deviation).

Characteristics	Subjects Who Underwent COVID-19 *n* = 33	Healthy Controls *n* = 35	*p*-Value
Age, years			
- Mean ± SD (range)	40.0 ± 12.8 (22–71)	38.9 ± 14.4 (21–61)	0.7
- Median	39	35	
Gender, *n* (*%*)			
- Male	6	9	0.45
- Female	27	26	
Height, cm			
- Mean ± SD (range)	167.1 ± 6.8 (152–180)	167.4 ± 8.6 (155–190)	0.75
- Median	168	165	
Weight, kg			
- Mean ± SD (range)	68.6 ± 16.4 (45–121)	68.4 ± 16.3 (46–110)	0.99
- Median	62	65	
Symptoms of COVID-19, *n* (%)		-	-
- Malaise	26 (78.8)		
- Anosmia/hyposmia	24 (72.7)		
- Arthralgia/myalgia	21 (63.6)		
- Headache	21 (63.6)		
- Fever	16 (48.5)		
- Cough	16 (48.5)		
- Rhinitis	13 (39.4)		
- Sore throat	11 (33.3)		
- Chest pain	10 (30.3)		
- Chills	10 (30.3)		
- Diarrhea	8 (24.2)		
- Dyspnea/shortness of breath	6 (18.2)		
- Nausea/vomiting	6 (18.2)		
- Conjunctivitis	1 (3.0)		
- Skin lesions	0 (0)		

**Table 2 medicina-58-00305-t002:** Comparison of the stabilographic parameters (opened eyes) in subjects who reported respiratory problems (*n* = 6) in the course of COVID-19 and those who did not suffer from dyspnea during the infection (*n* = 27) (AP, anterio-posterior direction; ML, medio-lateral direction; COP, center of pressure; ra, range; rms, root mean square; len, length of trajectory; v, velocity; RAMB, rambling; TREMB, trembling) (* statistically significant).

	Sagittal Plane (AP)				Frontal Plane (ML)			
	COVID-19 with Respiratory Problems *n* = 6	COVID-19 without Respiratory Problems *n* = 27			COVID-19 with Respiratory Problems *n* = 6	COVID-19 without Respiratory Problems *n* = 27		
Parameter	Median (Min–Max)	Median (Min–Max)	*p*	Effect Size	Median (Min–Max)	Median (Min–Max)	*p*	Effect Size
Entropy COP	0.09 (0.04–0.17)	0.07 (0.04–0.13)	0.234	0.432	0.09 (0.05–0.16)	0.06 (0.04–0.11)	0.098	0.612
raCOP, cm	2.15 (1.21–2.99)	1.77 (1.05–3.29)	0.234	0.432	1.68 (0.79–2.90)	1.68 (0.69–3.11)	0.981	0.016
rmsCOP, cm	0.38 (0.23–0.62)	0.38 (0.21–065)	0.797	0.098	0.33 (0.15–0.51)	0.34 (0.15–0.59)	0.944	0.033
lenCOP, cm	27.49 (18.24–37.32)	19.03 (15.56–35.40)	0.042 *	0.766	19.98 (15.67–29.14)	17.35 (8.94–29.9)	0.216	0.450
vCOP, cm/s	0.93 (0.62–1.26)	0.64 (0.53–1.20)	0.042 *	0.766	0.68 (0.53–0.99)	0.59 (0.30–1.02)	0.216	0.450
raRAMB, cm	1.99 (1.15–2.71)	1.69 (1.06–3.11)	0.363	0.329	1.54 (0.79–2.49)	1.61 (0.67–2.97)	0.981	0.016
rmsRAMB, cm	0.37 (0.22–0.57)	0.37 (0.21–0.64)	0.981	0.016	0.31 (0.15–0.47)	0.33 (0.14–0.56)	0.907	0.049
lenRAMB, cm	23.43 (17.02–31.21)	17.70 (14.60–30.65)	0.038 *	0.786	17.47 (14.81–26.13)	16.46 (8.16–26.18)	0.253	0.415
vRAMB, cm/s	0.79 (0.58–1.06)	0.60 (0.49–1.04)	0.038 *	0.786	0.59 (0.50–0.89)	0.56 (0.28–0.89)	0.253	0.415
raTREMB, cm	0.59 (0.21–1.21)	0.26 (0.12–0.87)	0.030 *	0.826	0.47 (0.18–0.56)	0.28 (0.08–0.66)	0.169	0.503
rmsTREMB, cm	0.05 (0.01–0.11)	0.02 (0.01–0.08)	0.038 *	0.786	0.04 (0.01 -0.05)	0.02 (0.00–0.06)	0.129	0.557
lenTREMB, cm	9.77 (4.33–16.17)	4.86 (2.57–14.52)	0.042 *	0.766	5.39 (2.89–7.85)	3.75 (1.46–9.34)	0.141	0.539
vTREMB, cm/s	0.33 (0.15–0.55)	0.16 (0.09–0.49)	0.042 *	0.766	0.18 (0.10–0.27)	0.13 (0.05–0.32)	0.141	0.539

**Table 3 medicina-58-00305-t003:** Comparison of the stabilographic parameters (closed eyes) in subjects who reported respiratory problems (*n* = 6) in the course of COVID-19 and those who did not suffer from dyspnea during the infection (*n* = 27) (AP, anterio-posterior direction; ML, medio-lateral direction; COP, center of pressure; ra, range; rms, root mean square; len, length of trajectory; v, velocity; RAMB, rambling; TREMB, trembling) (* statistically significant).

	Sagittal Plane (AP)				Frontal Plane (ML)			
	COVID-19 with Respiratory Problems *n* = 6	COVID-19 without Respiratory Problems *n* = 27			COVID-19 with Respiratory Problems *n* = 6	COVID-19 without Respiratory Problems *n* = 27		
Parameter	Median (Min–Max)	Median (Min–Max)	*p*	Effect Size	Median (Min–Max)	Median (Min–Max)	*p*	Effect Size
Entropy COP	0.12 (0.05–0.17)	0.07 (0.03–0.13)	0.038 *	0.786	0.08 (0.05–0.18)	0.07 (0.03–0.14)	0.591	0.196
raCOP, cm	3.23 (2.24–4.32)	2.35 (1.20–4.85)	0.053	0.726	2.79 (1.37–4.00)	2.22 (0.74–4.08)	0.316	0.363
rmsCOP, cm	0.61 (0.43–0.84)	0.46 (0.23–0.90)	0.141	0.539	0.56 (0.23–0.72)	0.44 (0.15–0.80)	0.316	0.363
lenCOP, cm	51.69 (29.41–62.82)	26.75 (19.53–50.30)	0.001 *	1.383	27.67 (20.11–62.02)	24.42 (10.79–45.93)	0.072	0.668
vCOP, cm/s	1.75 (1.00–2.13)	0.91 (0.66–1.70)	0.001 *	1.383	0.94 (0.68–2.10)	0.83 (0.37–1.56)	0.072	0.668
raRAMB, cm	2.79 (2.12–3.79)	2.22 (1.14–5.25)	0.118	0.575	2.43 (1.27–3.45)	2.04 (0.71–3.95)	0.469	0.262
rmsRAMB, cm	0.57 (0.39–0.78)	0.45 (0.21–0.90)	0.155	0.521	0.53 (0.20–0.67)	0.41 (0.14–0.76)	0.363	0.329
lenRAMB, cm	41.70 (25.07–45.69)	24.42 (17.82–38.70)	0.001 *	1.354	23.47 (18.24–49.60)	21.86 (9.69–37.99)	0.129	0.557
vRAMB, cm/s	1.41 (0.85–1.55)	0.83 (0.60–1.31)	0.001 *	1.354	0.80 (0.62–1.68)	0.74 (0.33–1.29)	0.129	0.557
raTREMB, cm	1.25 0.77–1.74)	0.47 (0.20–2.31)	0.002 *	1.271	0.76 (0.32–1.30)	0.43 (0.09–1.69)	0.047 *	0.746
rmsTREMB, cm	0.11 (0.06–0.20)	0.04 (0.01–0.21)	0.002 *	1.271	0.07 (0.02–0.14)	0.03 (0.00–0.18)	0.042 *	0.766
lenTREMB, cm	23.53 (10.05–35.43)	7.92 (3.75–30.06)	0.002 *	1.298	9.28 (5.51–27.90)	6.52 (1.77–24.01)	0.053	0.726
vTREMB, cm/s	0.80 (0.34–1.20)	0.27 (0.13–1.02)	0.002 *	1.298	0.31 (0.19–0.95)	0.22 (0.06–0.81)	0.053	0.726

## Data Availability

The datasets generated during and/or analyzed during the current study are available from the corresponding author on reasonable request.

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
