# Peer review of "COVID-19 and Postural Control—A Stabilographic Study Using Rambling-Trembling Decomposition Method"

_medicina, 2022, doi:10.3390/medicina58020305_

Round 1
Reviewer 1 Report
The study aim was to evaluate the effect of COVID-19 on postural (orthostatic) control in subjects who have recently recovered from the infection vs. healthy control group (33 vs. 35 individuals). The used "rambling-trembling decomposition method" considered as valuable. Patients who expressed olfactory dysfunction and/or respiratory distress because of COVID-19 showed symptoms of balance deficiency following COVID-19. Monitoring for neurological sequels and complications of COVID-19 should be evaluated and considered. I believe that this paper undoubtedly contributes to a better knowledge not only of the consequences and complications of COVID-19, but also their prediction - based on specific clinical abnormalities and/or symptoms.
Author Response
We are grateful to the reviewer for the very positive comments.
Reviewer 2 Report
The paper is well presented in all aspects of form and content. All sections are correct, and the data and results are presented in an appropriate and meaningful way.
Discussion is made on the results of the study, but it is too long in the first paragraphs (lines 222-269). Although the data included could be related to the contributions of the study, it could be shortened.
There is a typographical mistake in reference 12 (line 134).
Author Response
We are grateful to the reviewer for highly valuable comments. As suggested we have shortened the discussion and corrected the typo by ref. 12.